# Systemic Lupus Erythematosus in Hemodialysis: Survival Comparison and Mortality-Related Factors

**DOI:** 10.3390/medicina61010155

**Published:** 2025-01-17

**Authors:** Hwajeong Lee, Seok-Hui Kang

**Affiliations:** 1Division of Rheumatology, Department of Internal Medicine, Daegu Catholic University School of Medicine, Daegu 42472, Republic of Korea; hlee@cu.ac.kr; 2Division of Nephrology, Department of Internal Medicine, College of Medicine, Yeungnam University, Daegu 42415, Republic of Korea

**Keywords:** systemic lupus erythematosus, hemodialysis, mortality

## Abstract

*Background and Objectives*: We aimed to evaluate the outcome of systemic lupus erythematosus (SLE) patients undergoing hemodialysis (HD) by comparing the survival among HD patients with SLE, diabetes mellitus (DM), or other diseases in the Korean population. We also analyzed the factors affecting the survival of SLE patients undergoing HD. *Materials and Methods*: This retrospective study analyzed laboratory data from a national HD quality assessment program and claims data. The programs included maintenance HD patients aged ≥18 years. The patients were divided into three groups according to underlying comorbidities as follows: SLE (*n* = 569), DM (*n* = 24,665), and others (*n* = 31,004). *Results*: The Kaplan–Meier curve showed that the SLE group had better survival than the other two groups. Multivariate analysis showed that the hazard ratios were 0.89 (*p* = 0.334) in the others group and 1.32 (*p* = 0.015) in the DM group compared to those in the SLE group. In the SLE group, age, the Charlson Comorbidity Index (CCI) score, hemoglobin, serum creatinine levels, and systolic blood pressure (SBP) were associated with patient survival. *Conclusions*: This study showed that HD patients with SLE had better patient survival than those with DM and comparable survival with those with other diseases, except SLE or DM. In addition, age, CCI score, hemoglobin, serum creatinine level, and SBP were associated with survival in HD patients with SLE.

## 1. Introduction

End-stage renal disease (ESRD) is a chronic disease requiring renal replacement therapies. Among three renal replacement therapies (hemodialysis [HD], peritoneal dialysis, and kidney transplantation), HD is the most commonly used modality, and previous studies from the Republic of Korea and USA have shown that HD rates are approximately 81.0% and 61.7% in Korea and the USA, respectively [1,2]. HD patients are associated with poor outcomes compared to those of the general population, and efforts to evaluate factors of mortality in HD patients are ongoing.

Two metabolic diseases, diabetes mellitus (DM) and hypertension, are the most common causes of ESRD [1]. Therefore, most previous studies used cohorts that included patients with these two diseases or only used patients with these diseases. Studies on ESRD patients with rare diseases have been relatively less common than those on patients with DM or hypertension.

Systemic lupus erythematosus (SLE) is a chronic autoimmune disease that involves multiple organs. SLE can also affect the kidneys, and it may be associated with the development of ESRD through the progression of lupus nephritis, or in combination with underlying comorbidities. Nissenson et al. showed that SLE had similar outcomes to those of ESRD patients with other connective tissue diseases and better outcomes compared to those of ESRD patients with DM [3]. A study using the US Renal Data System dataset showed that ESRD patients with SLE had a greater survival rate than patients with DM [4]. In addition, female ESRD patients with lupus nephritis had a similar rate of hospitalization for acute myocardial infarction and cerebrovascular accident compared to those with ESRD from DM. A study from the French registry showed similar results to those from the US Renal Data System [5]. Cardiovascular and all-cause mortality in patients with SLE was lower than in patients with DM but three times higher than in patients with polycystic kidney disease. Although some studies have evaluated the survival or survival-related factors in ESRD patients with SLE compared to those with other diseases, data on these issues have been insufficient. Further, dialysis modality, region, and ethnicity may have affected these results.

Patients with SLE undergoing HD face a unique set of challenges due to immune system abnormalities caused by SLE, as well as due to cardiovascular and metabolic problems associated with HD. Traditionally, the management of SLE has focused on controlling disease activity, stabilizing the immune system, and preventing or treating disease flares. However, patients undergoing HD experience additional immune suppression due to renal failure, which sets them apart from general patients with SLE. In these patients, both SLE-specific and HD-specific factors, such as dialysis adequacy and chronic comorbidities including DM and hypertension, are likely to have a significant impact. Despite this, the analysis of HD-associated factors in patients with SLE undergoing HD is limited due to the small number of such cases. Therefore, investigating the use of immunosuppressive agents (ISAs) and the impact of HD-associated factors on patient outcomes and comparing these findings with other disease groups may enhance our understanding of this unique population of patients with SLE undergoing HD. Therefore, in this study, we aimed to evaluate the outcome of SLE patients undergoing HD by comparing the survival among HD patients with SLE, DM, and other diseases in the Korean population. We also analyzed the factors affecting the survival of SLE patients undergoing HD.

## 2. Materials and Methods

### 2.1. Data Source and Study Population

This retrospective study analyzed laboratory data from a national HD quality assessment program and the claims data from the Health Insurance Review and Assessment (HIRA) of the Republic of Korea [6,7]. Briefly, the fourth, fifth, and sixth HD quality assessment programs were performed between July 2013 and December 2013; between July 2015 and December 2015; and between March 2018 and August 2018, respectively. The programs included maintenance HD patients (≥3 months); undergoing HD at least twice a week (≥8 per month); and aged ≥18 years.

We also analyzed the claims data of all HD patients who had undergone HD quality assessment from the HIRA. The Korean National Healthcare System and the Medical Aid program cover almost the entire South Korean population. The HIRA, as a government-affiliated organization, has nearly all medical information of patients, from diagnoses and past medical records to procedural data.

The study was approved by the institutional review board (IRB) of the Yeungnam University Medical Center (approval no. YUMC 2022-01-010). Informed consent was not obtained from the patients since the records and information of the participants were anonymized and de-identified before the analysis. The IRB also waived the need for obtaining informed consent. The study was conducted ethically in accordance with the World Medical Association Declaration of Helsinki.

### 2.2. Study Population and Variables

HD quality assessment data were collected using a web-based data collection system. The data collected included age (years), sex, post-dialysis body weight (kg) as an indicator of dry weight, body mass index (kg/m^2^), underlying disease of ESRD, HD vintages (days), and type of vascular access. Laboratory data from the assessment included hemoglobin (g/dL), Kt/V_urea_, serum albumin (g/dL), serum calcium (mg/dL), serum phosphorus (mg/dL), pre-dialysis systolic blood pressure (SBP, mmHg), pre-dialysis diastolic blood pressure (DBP, mmHg), and the ultrafiltration volume (UFV, L/session) as an indicator of weight gain. These data were collected monthly, and all laboratory values were averaged from the monthly collected values. Kt/V_urea_ was calculated using Daugirdas’ equation [8].

The patients were divided into three groups according to underlying comorbidities as follows: SLE, DM, and others. Underlying disease of ESRD from the HD quality assessment data was described as one among DM, hypertension, chronic glomerulonephritis, or others. Therefore, we did not evaluate SLE using the HD quality assessment data, and SLE was evaluated using the claims data for a year before the evaluation of the HD quality assessment program. First, the SLE group was defined using the International Classification of Diseases, 10th revision, Clinical Modification (ICD-10) system of M32. Second, the DM group was defined as that which had DM as an underlying disease of ESRD during the HD quality assessment. Third, the others group was defined as that with patients of other diseases, except SLE or DM.

The use of medications was defined as one or more prescriptions of a drug during a six-month period of HD quality assessment. The impact of medications for lupus (hydroxychloroquine, steroids [prednisolone, methylprednisolone, deflazacort], azathioprine, mycophenolate mofetil, mycophenolic acid, cyclophosphamide, calcineurin inhibitor [tacrolimus, cyclosporine], rituximab, or belimumab), renin–angiotensin system blockades (RASBs), or anti-hypertensive drugs including RASB were evaluated.

The presence of other comorbidities was evaluated for a year before the HD quality assessment and was defined using the codes utilized by Quan et al. [9,10]. Finally, the Charlson Comorbidity Index (CCI) score was calculated. During the follow-up, clinical outcomes other than death were defined using the Electronic Data. The codes were as follows: O7072, O7071, and O7061 for peritoneal dialysis and R3280 for kidney transplantation.

### 2.3. Statistical Analyses

Data were analyzed using the SAS Enterprise Guide version 7.1 (SAS Institute, Cary, NC, USA) or R version 3.5.1 (R Foundation for Statistical Computing, Vienna, Austria). Categorical variables were presented as numbers and percentages, whereas continuous variables were presented as the means ± standard deviations. Pearson’s χ^2^ test or Fisher’s exact test was used to analyze the categorical variables. For continuous variables, the means were compared using a one-way analysis of variance, followed by the Tukey post hoc test. The survival estimates were calculated using the Kaplan–Meier curve and Cox regression analyses. *p*-values for the comparison of survival curves were determined using the log-rank test. Multivariate Cox regression analyses were adjusted for age, sex, CCI score, HD vintage, UFV, Kt/V_urea_, hemoglobin, serum albumin, serum creatinine, serum phosphorus, serum calcium, SBP, and DBP and were performed using the backward mode. Statistical significance was set at *p* < 0.05.

## 3. Results

### 3.1. Participant Clinical Characteristics

The numbers of patients included in the fourth, fifth, and sixth HD quality assessment program were 21,846, 35,538, and 31,294, respectively. Among these, we excluded repeated participants or participants with an insufficient dataset (75 in the fourth, 13,795 in the fifth, and 18,570 in the sixth). Finally, 56,238 were included in our study. The numbers of patients with SLE, DM, and other diseases were 569, 24,665, and 31,004, respectively. Baseline characteristics are shown in Table 1.

The SLE group had the highest Kt/V_urea_ and DBP among the three groups. The DM group had the highest age, proportion of male sex, CCI score, UFV, and SBP and the lowest serum calcium and phosphorus levels among the three groups. The others group had the highest HD vintages, follow-up durations, hemoglobin, and serum albumin levels among the three groups. For weekly HD sessions, the proportion of patients undergoing HD twice per week was 48 (8.5%) in the SLE group; 2074 (8.4%) in the DM group; and 3613 (11.7%) in the others group. Although twice-weekly HD sessions were slightly more common in the others group, the majority of patients in all groups underwent HD thrice per week.

The post-dialysis body weights in the SLE, DM, and others groups were 57.4 ± 12.5, 62.7 ± 11.6, and 59.3 ± 11.8 kg, respectively (*p* < 0.001). Overall, 481 (84.5%), 22,001 (89.2%), and 24,493 (79.0%) patients used anti-hypertensive drugs in the SLE, DM, and others groups, respectively. The three groups had similar UFV percentages relative to the post-dialysis body weight (approximately 3.7% across the groups). However, the DM group exhibited the highest pre-dialysis SBP and the highest proportion of anti-hypertensive drug or RASB use among the three groups. These results suggest that the higher use of anti-hypertensive drugs, particularly RASBs in patients with DM, may be attributable to a higher prevalence of cardiovascular diseases in this group. Additionally, the elevated pre-dialysis SBP in the DM group may indicate an effort to mitigate the risk of intradialytic hypotension compared to other groups. However, differences in SBP and the use of anti-hypertensive drugs between the SLE and others groups were not as pronounced as those observed in the DM group.

Although our study did not directly access disease activity, we evaluated medications for SLE. The number of patients who used steroids, hydroxychloroquine, mycophenolate mofetil or mycophenolic acid, cyclophosphamide, azathioprine, and calcineurin inhibitors was 250 (43.9%), 84 (14.8%), 33 (5.8%), 5 (0.9%), 9 (1.6%), and 22 (3.9%), respectively. However, the number of patients who did not use any of these drugs was 300 (52.7%). Compared to the results of a previous study, the proportion of patients with SLE undergoing HD who used ISAs was lower [11]. This may indicate that patients with SLE undergoing have relatively lower disease activity than those without HD.

### 3.2. Survival Analysis

The number of deaths at the end of follow-up was 133 (23.4%) in the SLE group, 12,083 (49.0%) in the DM group, and 9865 (31.8%) in the others group (Table 2).

The SLE group had greater proportions of survival or transfer to another modality than the other two groups. The Kaplan–Meier curve showed that the SLE group had better survival than the other two groups (Figure 1).

We performed subgroup analyses according to age and sex (Figure 2 and Appendix A).

These showed similar trends to those from the total number of patients, but statistical significance between the SLE and the other groups was not obtained.

Univariate Cox regression analysis showed that the hazard ratios were 1.24 (95% CI, 1.04–1.47; *p* < 0.001) in the others group and 2.24 (95% CI, 1.89–2.65; *p* < 0.001) in the DM group compared to those in the SLE group. Multivariate analysis showed that the hazard ratios were 0.89 (95% CI, 0.71–1.12; *p* = 0.334) in the others group and 1.32 (95% CI, 1.06–1.66; *p* = 0.015) in the DM group compared to those in the SLE group (Table 3).

The SLE group showed comparable survival to that of the others group and better survival than that of the DM group.

In the SLE group, age, CCI score, hemoglobin, serum creatinine levels, and SBP were associated with patient survival (Table 4).

Age was a significant factor, with patients aged ≥65 years having a much higher risk of mortality (HR = 2.57, 95% CI: 1.43–4.61, *p* = 0.002). A higher CCI score was associated with increased mortality (HR = 1.17 per 1 score increase, 95% CI: 1.07–1.28, *p* < 0.001). These variables were similar to those of the DM and others groups (Appendix A). In the SLE group, a hemoglobin level < 11 g/dL was associated with higher mortality. Patients with SBP < 140 mmHg had better survival and those with SBP > 140 mmHg had poorer survival than those with the reference standard of 140 mmHg (Figure 3).

Patients with serum creatinine <10 mg/dL had poorer survival and those with serum creatinine >10 mg/dL had better survival than those with the reference standard of 10 mg/dL.

### 3.3. Subgroup Analysis

We compared Kt/V_urea_, UFV, and vascular access between SLE and glomerulonephritis (GN). A total of 7183 patients had GN, among whom, 6335 (88.2%), 717 (10.0%), and 131 (1.8%) patients had vascular access through an arteriovenous fistula, arteriovenous graft, or catheter, respectively. The Kt/V_urea_ and UFV was 1.57 ± 0.32 and 2.17 ± 0.96 L/session, respectively. The proportions of arteriovenous grafts and Kt/V_urea_ were greater in patients with SLE than in those with GN (*p* < 0.001 for two variables). UFVs were similar between the two groups (*p* = 0.935).

In addition, we evaluated the effect of DM or hypertension in patients with SLE undergoing HD. The number of patients with SLE alone, both SLE and DM, and DM alone was 428, 141, and 24,524, respectively. The number of patients with SLE alone, both SLE and hypertension, and hypertension alone was 115, 481, and 46,590, respectively. The proportion of patients with DM or hypertension among those with SLE was 24.8% and 80.7%, respectively. Appendix A shows Cox regression analyses based on SLE and DM or hypertension. Patients with SLE and concurrent DM were associated with a high mortality rate, with DM exhibiting a higher mortality rate than SLE alone. However, the presence of hypertension, whether as a comorbidity or as an independent condition, was not associated with mortality in patients with SLE.

We divided patients with SLE into two groups: those who used and those who did not use ISAs. Multivariate Cox regression showed that SLE not treated with ISAs demonstrated better patient survival than that observed in the DM group (Appendix A). However, SLE treated with ISAs exhibited poorer patient survival than that of the others group and similar patient survival compared with the DM group. We revealed that patients with SLE had a different patient survival profile, based on the use of ISAs.

## 4. Discussion

This study included 56,238 patients undergoing HD who participated in a HD quality assessment program. Survival analysis showed that the SLE group had better survival than the other two groups, and the subgroup analyses according to age and sex showed similar trends to those from the patients as a whole. Multivariate Cox regression analyses showed that the SLE group had comparable survival with the others group and better survival than those in the DM group.

We performed a risk factor analysis for mortality using patients divided by groups. Age, CCI score, hemoglobin, serum creatinine, and SBP were associated with mortality in the SLE group. Spline curves also showed that hemoglobin < 11 g/dL, SBP > 140 mmHg, and serum creatinine < 10 mg/dL were associated with higher mortality in the SLE group.

Previous studies have shown the effect of SLE on the outcomes of patients with ESRD. Hellerstedt et al. evaluated 2728 patients with ESRD and showed that patients with focal segmental glomerulosclerosis or SLE had better patient survival than those with DM or malignancy, but the study evaluated all patients with ESRD, including those with kidney transplantation [12]. Two studies have evaluated the USRDS dataset, and Ward showed that the overall death rate was greater in SLE than in others, but better in SLE than in DM [4,13]. O’Shaughnessy et al. analyzed the outcomes in patients with GN using the USRDS data and found that patients with SLE had poorer outcomes than those with IgA nephropathy; comparable outcomes to those with focal segmental glomerulosclerosis; and better outcomes than those with membranoproliferative glomerulonephritis, membranous nephropathy, or vasculitis [13]. Levy et al. evaluated the French registry and showed higher rates of cardiovascular deaths in patients with SLE than in those of matched patients with polycystic kidney disease [5]. Three studies have evaluated Asian populations with ESRD (one study each from Japan, China, and Taiwan) [14,15,16]. Two studies from Japan and China have shown higher rates of patient deaths in those with SLE than in those without SLE, but one study from Taiwan showed similar outcomes between patients with SLE and those without SLE and DM. Previous studies have revealed that patients with SLE had inconsistent results compared to those with other diseases, excluding DM, but patients with SLE had better outcomes than those with DM. These results were similar to those from our study.

The strength of the present study is the large sample size using the claims data and analyses of some laboratory data beyond simple claims data. This study included a relatively large sample of patients with SLE. In addition, this study evaluated some laboratory data including prognostic indicators, such as Kt/V_urea_ or UFV in HD patients, in addition to analyses using the claims data. These laboratory data were collected during the HD quality assessment. The program was used to evaluate the quality of the HD facility and graded to a 5-star rating [17]. It was used to present different reimbursements for each HD facility. Therefore, the quality of the data garnered for patients’ characteristics and laboratory studies may have been high.

This study also evaluated mortality-related risk factors in HD patients with SLE. The importance of comorbidities in HD patients with SLE has been evaluated in some studies [5,16,18]. Old age and comorbidities are associated with mortality in HD patients with SLE, and without SLE. A study using a Brazilian cohort showed SLE activity as a risk factor for mortality in HD patients with SLE [19]. Two studies have shown poor outcomes in male patients than in female patients, but the present study did not show significant differences in mortality according to sex [16,20]. In the present study, hemoglobin and creatinine levels were inversely associated with mortality and positively associated with SBP. Hemoglobin, serum creatinine levels, and SBP are well-known factors associated with prognosis in HD patients, regardless of underlying diseases. In this study, no significant associations were found between mortality and other factors, such as nutritional indicators or dialysis adequacy. This may be due to two factors: a relatively small sample size of HD patients with SLE (*n* = 569) and adequate levels of various mortality-associated indicators in most of our HD patients [21]. In the sixth HD quality assessment program, 95.6% of all HD patients had an adequate dialysis dose, and the hemoglobin level > 10 g/dL was 87.1%. The high quality of the HD facility and low proportion of patients with inadequate indicators may attenuate the importance of some of the indicators for mortality in HD patients.

Our dataset originated from the HD quality assessment program and its claims data. The purpose of the program was to collect factors associated with improved patient outcomes from a center-based perspective, rather than patient-specific factors linked to individual prognoses. These factors included calcium and phosphorus control, anemia management, and dialysis adequacy. Additionally, due to the nationwide nature of this study, factors that were burdensome to collect or prone to error were excluded. Consequently, data on residual renal function, which requires 24 h urine collection and is both inconvenient and prone to inaccuracies, were not collected and analyzed in this study. However, most patients included in this study had undergone HD for an average of more than three years. Previous data from a study using the DOPPS cohort showed that 76.1% of patients undergoing HD for more than one year had a urine volume of less than 200 mL/day, and this percentage increased to 83.7% for those undergoing HD for over three years [22]. Furthermore, Japanese patients undergoing HD lost 86.0% of residual renal function after one year and 90.4% after three years. Considering these findings, it is likely that most patients included in our study had minimal or no urine volume.

In addition, our study did not include detailed data on HD sessions, such as changes in blood pressure during HD, HD modalities, blood flow rate during HD, vascular access pressures, dialyzer types, and the duration of HD sessions, which may have served as additional confounding factors affecting patient outcomes. However, in population-based registry data, some information on HD sessions is available [1]. In South Korea, the use of synthetic membranes, primarily polysulfone, has been increasing and exceeded 90% by 2009. Additionally, in South Korea, reimbursements for HD sessions are provided uniformly, regardless of the dialyzer type or dialysis modality. As a result, high-flux dialyzers and HD are predominantly used, while medium-cutoff dialyzers or hemodiafiltration are available for only a limited number of patients. In fact, data from 2018 indicate that only 17% of patients underwent hemodiafiltration at least once per week, and this percentage decreased to 11.1% by 2022, suggesting that the vast majority of HD modalities remain conventional HD. Furthermore, over 80% of patients use dialyzers with surface areas between 1.0 and 2.0 m^2^.

This study has some limitations. First, it was a retrospective study that analyzed the dataset during the HD quality assessment period and the claims data without a medical chart review. Second, the causal relationship between ESRD and SLE was unclear. Our study did not distinguish between ESRD development by progression of lupus nephritis and ESRD development regardless of SLE. These two conditions may have been different in terms of clinical outcomes. Third, as Ribeiro et al. reported, high non-renal SLE disease activity (nrSLEDAI) is strongly associated with five-year mortality in lupus patients on dialysis [19]. However, our study did not include data on lupus disease activity, so we could not assess its impact on the prognosis of HD patients with SLE.

## 5. Conclusions

This study showed that HD patients with SLE had better patient survival rates than those with DM and comparable survival to those with other diseases except for SLE and DM. In addition, age, CCI score, hemoglobin and serum creatinine levels, and SBP were associated with survival in HD patients with SLE.

## Figures and Tables

**Figure 1 medicina-61-00155-f001:**
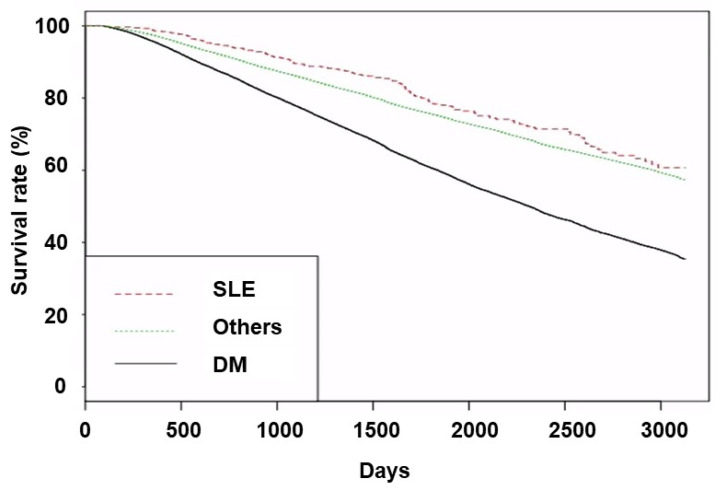
Kaplan–Meier curves of patient survival according to underlying disease. *p*-values were <0.001 for trend, <0.001 for DM vs. SLE or others, and 0.013 for SLE vs. others. Abbreviations: SLE, systemic lupus erythematosus; DM, diabetes mellitus.

**Figure 2 medicina-61-00155-f002:**
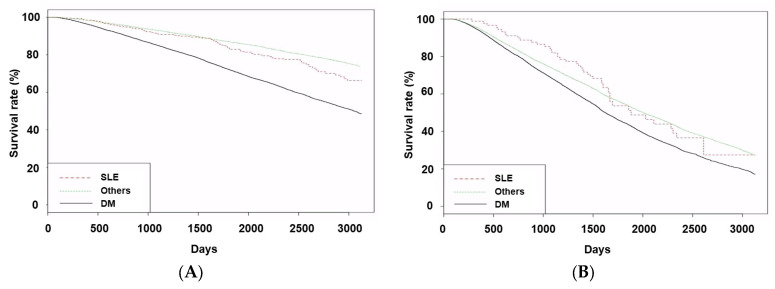
Kaplan–Meier curves of patient survival were divided into young (aged < 65 years, (**A**)) and old (aged ≥ 65 years, (**B**)) patients. For young patients, *p*-values were <0.001 for trend, <0.001 for DM vs. SLE or others, and 0.057 for SLE vs. others. For old patients, *p*-values were <0.001 for the trend, 0.025 for DM vs. SLE, <0.001 for DM vs. others, and 0.572 for SLE vs. others. Abbreviations: SLE, systemic lupus erythematosus; DM, diabetes mellitus.

**Figure 3 medicina-61-00155-f003:**
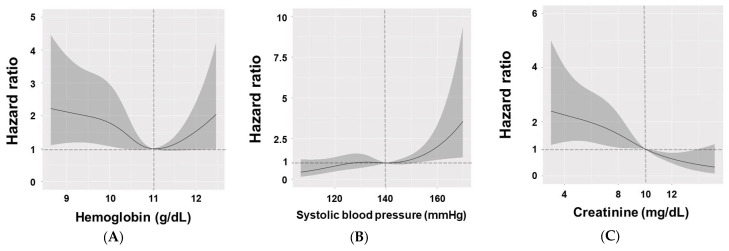
Hazard ratios of hemoglobin (**A**), systolic blood pressure (**B**), and serum creatinine levels (**C**) in patients with SLE. The graphs were expressed with 95% confidence intervals (gray) and reference line (11 g/dL for hemoglobin, 140 mmHg for the systolic blood pressure, 10 mg/dL for serum creatinine, and 1.0 for the hazard ratio).

**Table 1 medicina-61-00155-t001:** Patient clinical characteristics.

	SLE (*n* = 569)	DM (*n* = 24,665)	Others (*n* = 31,004)	*p*
Age (years)	50.3 ± 13.5	62.3 ± 11.4 *	58.8 ± 14.0 *^†^	<0.001
Sex (male, %)	173 (30.4%)	15,267 (61.9%)	9398 (52.2%)	<0.001
Hemodialysis vintage (days)	1515 ± 1699	1160 ± 1135 *	1874 ± 195 *^†^	<0.001
CCI score	8.0 ± 2.8	8.8 ± 2.4 *	6.5 ± 2.9 *^†^	<0.001
Follow-up duration (days)	1817 ± 795	1717 ± 839 *	1952 ± 880 *^†^	<0.001
Type of vascular access				<0.001
Arteriovenous fistula	453 (79.6%)	19,935 (80.8%)	26,439 (85.3%)	
Arteriovenous graft	93 (16.3%)	4146 (16.8%)	3856 (12.4%)	
Catheter	23 (4.0%)	584 (2.4%)	709 (2.3%)	
Kt/V_urea_	1.67 ± 0.33	1.49 ± 0.26 *	1.55 ± 0.28 *^†^	<0.001
Ultrafiltration volume (L/session)	2.15 ± 0.93	2.35 ± 0.91 *	2.20 ± 0.99 ^†^	<0.001
Post-dialysis body weight (kg)	57.4 ± 12.5	62.7 ± 11.6 *	59.3 ± 11.8 *^†^	<0.001
Body mass index (kg/m^2^)	21.7 ± 3.5	23.3 ± 3.5 *	22.1 ± 3.4 *^†^	<0.001
Hemoglobin (g/dL)	10.6 ± 0.8	10.6 ± 0.7	10.7 ± 0.9 *^†^	<0.001
Serum albumin (g/dL)	3.96 ± 0.34	3.95 ± 0.36	4.00 ± 0.34 *^†^	<0.001
Serum phosphorus (mg/dL)	5.2 ± 1.4	4.8 ± 1.3 *	5.1 ± 1.4 ^†^	<0.001
Serum calcium (mg/dL)	8.9 ± 0.8	8.8 ± 0.8 *	9.0 ± 0.8 ^†^	<0.001
Systolic blood pressure (mmHg)	137 ± 16	145 ± 15 *	139 ± 15 *^†^	<0.001
Diastolic blood pressure (mmHg)	80 ± 9	77 ± 10 *	79 ± 9 *^†^	<0.001
Use of RASB	360 (63.2%)	17,759 (72.0%)	18,509 (59.7%)	<0.001
Use of anti-hypertensive drugs	481 (84.5%)	22,001 (89.2%)	24,493 (79.0%)	<0.001
Number of HD sessions				<0.001
Twice per week	48 (8.5%)	2074 (8.4%)	3613 (11.7%)	
Thrice per week	521 (91.5%)	22,591 (91.6%)	27,391 (88.3%)	
Medications for lupus				
Hydroxychloroquine	84 (14.8%)	-	-	
Steroid	250 (43.9%)	-	-	
MMF or MPA	33 (5.8%)	-	-	
Cyclophosphamide	5 (0.9%)	-	-	
Azathioprine	9 (1.6%)	-	-	
Rituximab	0	-	-	
Belimumab	0	-	-	
Calcineurin inhibitors	22 (3.9%)	-	-	

Data are expressed as the means ± standard deviations for continuous variables and as numbers (percentages) for categorical variables. *p*-values are tested using one-way analysis of variance, followed by the Tukey post hoc test and Pearson’s χ^2^ test for categorical variables. Abbreviations: SLE, systemic lupus erythematosus; DM, diabetes mellitus; CCI, Charlson comorbidity index; MMF, mycophenolate mofetil; MPA, mycophenolic acid; RASB, renin–angiotensin system blockade. * *p* < 0.05 vs. SLE, ^†^ *p* < 0.05 vs. DM.

**Table 2 medicina-61-00155-t002:** Status of patients at the end-point of follow-up.

	SLE	DM	Others	*p*-Value
Survivor	351 (61.7%)	11,290 (45.8%)	18,057 (58.2%)	<0.001
Death	133 (23.4%)	12,083 (49.0%)	9865 (31.8%)	
Transfer to PD	14 (0.5%)	100 (0.4%)	105 (0.3%)	
Transfer to KT	71 (12.5%)	1192 (4.8%)	2977 (9.6%)	

*p*-values are tested using Pearson’s χ^2^ test. Abbreviations: SLE, systemic lupus erythematosus; DM, diabetes mellitus; PD, peritoneal dialysis; KT, kidney transplantation.

**Table 3 medicina-61-00155-t003:** Cox regression analyses for patient survival.

	Multivariate	
	HR (95% CI)	*p*
Disease group (ref: SLE)		
Others	0.89 (0.71–1.12)	0.334
DM	1.32 (1.06–1.66)	0.015
Age (ref: <65 years)	2.61 (2.51–2.71)	<0.001
Sex (ref: male)	0.72 (0.69-0.75)	<0.001
CCI score (per 1 score increase)	1.07 (1.06–1.08)	<0.001
Hemoglobin (per 1 g/dL increase)	0.89 (0.87–0.91)	<0.001
Serum creatinine (per 1 mg/dL increase)	0.92 (0.91–0.93)	<0.001
Systolic blood pressure (per 1 mmHg increase)	1.00 (1.00–1.01)	<0.001
Serum calcium (per 1 mg/dL increase)	1.07 (1.05–1.09)	<0.001
Serum phosphorus (per 1 mg/dL increase)	1.02 (1.00–1.03)	0.014
Kt/V_urea_ (per 1 unit increase)	0.89 (0.83–0.96)	0.003
Serum albumin (per 1 g/dL increase)	0.58 (0.55–0.61)	<0.001
Hemodialysis vintages (per 1 day increase)	1.00 (1.00–1.00)	<0.001
Ultrafiltration volume (per 1 L/session increase)	1.02 (1.00–1.04)	0.015

The multivariate analysis was adjusted for the disease group, age, sex, CCI score, hemodialysis vintage, ultrafiltration volume, Kt/V_urea_, hemoglobin, serum albumin, serum creatinine, serum phosphorus, serum calcium, systolic blood pressure, and diastolic blood pressure and was performed using the backward mode. Abbreviations: HR, hazard ratio; CI, confidence interval; SLE, systemic lupus erythematosus; DM, diabetes mellitus; CCI, Charlson comorbidity index.

**Table 4 medicina-61-00155-t004:** Cox regression analyses for survival in patients with SLE.

	Multivariate	
	HR (95% CI)	*p*
Age (ref: <65 years)	2.57 (1.43–4.61)	0.002
CCI score (per 1 score increase)	1.17 (1.07–1.28)	<0.001
Hemoglobin (per 1 g/dL increase)	0.74 (0.54–0.98)	0.035
Serum creatinine (per 1 mg/dL increase)	0.90 (0.83–0.98)	0.016
Systolic blood pressure (per 1 mmHg increase)	1.02 (1.01–1.04)	<0.001

The multivariate analysis was adjusted for the disease group, age, sex, CCI score, hemodialysis vintage, ultrafiltration volume, Kt/V_urea_, hemoglobin, serum albumin, serum creatinine, serum phosphorus, serum calcium, systolic blood pressure, and diastolic blood pressure and was performed using the backward mode. Abbreviations: HR, hazard ratio; CI, confidence interval; SLE, systemic lupus erythematosus; CCI, Charlson comorbidity index.

## Data Availability

The raw data were generated at the Health Insurance Review and Assessment Service. The database can be requested from the Health Insurance Review and Assessment Service by sending a study proposal including the purpose of the study, study design, and duration of analysis through an e-mail (turtle52@hira.or.kr) or at the web site (https://www.hira.or.kr: accessed on 17 January 2025). The authors cannot distribute the data without permission.

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
