# Peer review of "Systemic Lupus Erythematosus in Hemodialysis: Survival Comparison and Mortality-Related Factors"

_medicina, 2025, doi:10.3390/medicina61010155_

Round 1
Reviewer 1 Report
Comments and Suggestions for Authors
The study aimed to evaluate the outcome of systemic lupus erythematosus (SLE) patients undergoing hemodialysis by comparing the survival among hemodialysis patients with SLE, diabetes mellitus (DM), or other diseases in Korea population.
The paper is interesting but several questions need to be raised
The introduction must be more extensive and aimed at the purpose.
The study has many methodological limitations regarding SLE disease activity. For this reason I suggest focusing the entire discussion on hemodialysis data, therefore KT/V and vascular access with respect to the differences between SLE and other glomerulonephrthis such ad FSG, membranoproliferative vasculitis in terms of outcome. How many patients with SLE had diabetes or hypertension? Have the authors investigated the different outcome of patients with SLE and diabetes or hypertension?
Author Response
The study aimed to evaluate the outcome of systemic lupus erythematosus (SLE) patients undergoing hemodialysis by comparing the survival among hemodialysis patients with SLE, diabetes mellitus (DM), or other diseases in Korea population.
The paper is interesting but several questions need to be raised
The introduction must be more extensive and aimed at the purpose.
Answer: Thank you for your comment. We have revised the Introduction section and added the following: Patients with SLE undergoing HD face a unique set of challenges due to immune system abnormalities caused by SLE, as well as due to cardiovascular and metabolic problems associated with HD. Traditionally, the management of SLE has focused on controlling disease activity, stabilizing the immune system, and preventing or treating disease flares. However, patients undergoing HD experience additional immune suppression due to renal failure, which sets them apart from general patients with SLE. In these patients, both SLE-specific and HD-specific factors, such as dialysis adequacy and chronic comorbidities including DM and hypertension, are likely to have a significant impact. Despite this, the analysis of HD-associated factors in patients with SLE undergoing HD is limited due to the small number of such cases. Therefore, investigating the use of immunosuppressive agents, the impact of HD-associated factors on patient outcomes, and comparing these finding with other disease group may enhance our understanding of this unique population of patients with SLE undergoing HD.
The study has many methodological limitations regarding SLE disease activity. For this reason I suggest focusing the entire discussion on hemodialysis data, therefore KT/V and vascular access with respect to the differences between SLE and other glomerulonephrthis such and FSG, membranoproliferative vasculitis in terms of outcome. How many patients with SLE had diabetes or hypertension? Have the authors investigated the different outcome of patients with SLE and diabetes or hypertension?
Answer: Thank you for your comment.
We compared Kt/Vurea, ultrafiltration volume, and vascular access between SLE and glomerulonephritis (GN). A total of 7,183 patients had GN, among whom 6,335 (88.2%), 717 (10.0%), and 131 (1.8%) patients had vascular access through an arteriovenous fistula, arteriovenous graft, or catheter, respectively. The Kt/Vurea and ultrafiltration volume were 1.57 ± 0.32, and 2.17 ± 0.96 L/session, respectively. The proportion of arteriovenous fistulas or catheters and Kt/Vurea were greater in patients with SLE than in those with GN (P < 0.001 for two variables). Ultrafiltration volumes were similar between the two groups (P = 0.935).
The use of medications was defined as one or more prescriptions of a drug during a six-month period of HD quality assessment. The impact of medications for lupus (hydroxychloroquine, steroids [prednisolone, methylprednisolone, deflazacort, azathioprine], mycophenolate mofetil, mycophenolic acid, cyclophosphamide, calcineurin inhibitor [tacrolimus, cyclosporine], rituximab, or belimumab), renin-angiotensin system blockades (RASBs), and anti-hypertensive drugs including RASB, were evaluated. Although our study did not directly access disease activity, we evaluated medications for SLE. The number of patients who used steroids, hydroxychloroquine, mycophenolate mofetil or mycophenolic acid, cyclophosphamide, azathioprine, and calcineurin inhibitors was 250 (43.9%), 84 (14.8%), 33 (5.8%), 5 (0.9%), 9 (1.6%), and 22 (3.9%), respectively. However, the number of patients who did not use any of these drugs was 300 (52.7%). Compared to the results of a previous study, the proportion of patients with SLE undergoing HD who did not use immunosuppressive agents (ISAs) was lower [1]. This may indicate that patients with SLE undergoing HD have relatively lower disease activity than those without HD.
In addition, we evaluated the effect of DM or hypertension in patients with SLE undergoing HD. The number of patients with SLE alone, both SLE and DM, and DM alone was 428, 141, and 24,524, respectively. The number of patients with SLE alone, both SLE and hypertension, and hypertension alone was 115, 481, and 46,590, respectively. The proportion of patients with DM or hypertension among those with SLE was 24.8% and 80.7%, respectively. Table S3 shows Cox regression analyses based on SLE and DM or hypertension.
Table S3. Cox regression analyses of patients with SLE based on SLE with diabetes or hypertension
|
|
Multivariate |
|
|
HR (95% CI) |
P |
|
|
Ref: SLE without diabetes |
|
|
|
SLE with diabetes |
2.00 (1.31–3.07) |
0.001 |
|
Diabetes alone |
1.59 (1.26–2.01) |
<0.001 |
|
Ref: SLE with diabetes |
|
|
|
Diabetes alone |
0.79 (0.55–1.14) |
0.207 |
|
Ref: SLE without hypertension |
|
|
|
SLE with hypertension |
0.98 (0.58–1.65) |
0.934 |
|
Hypertension alone |
1.01 (0.63–1.63) |
0.953 |
|
Ref: SLE with hypertension |
|
|
|
Hypertension alone |
1.04 (0.84–1.29) |
0.741 |
The multivariate analysis was adjusted for the disease group, age, sex, Charlson comorbidity index score, hemodialysis vintage, ultrafiltration volume, Kt/Vurea, systolic blood pressure, diastolic blood pressure, as well as levels of hemoglobin, serum albumin, serum creatinine, serum phosphorus, and serum calcium; and it was performed using the backward mode.
Abbreviations: CI, confidence interval; HR, hazard ratio; SLE, systemic lupus erythematosus
Patients with SLE and concurrent diabetes were associated with a high mortality rate, with DM exhibiting a higher mortality rate than SLE alone. However, the presence of hypertension, whether as a comorbidity or as an independent condition, was not associated with mortality in patients with SLE.
We have added these comments in the Methods and Results section.
Added reference
[1] Han JY, Cho SK, Kim H, Jeon Y, Kang G, Jung SY, Jang EJ, Sung YK. Increased cardiovascular risk in Korean patients with systemic lupus erythematosus: a population-based cohort study. Sci Rep. 2024 Jan 11;14(1):1082.

Reviewer 2 Report
Comments and Suggestions for Authors
First of all, we would like to thank the authors for their original and interesting work on the comparison of survival and mortality-related factors among patients undergoing hemodialysis for SLE, DM2 and other causes of CKD in a Korean population.
It is a very interesting manuscript, but it has an important gap on the SLE population studied, although it is commented in the limitations of the study, it would be interesting to describe in some way parameters that identify the severity, activity, time of disease and medication of SLE, probably adding these data in the baseline characteristics of this group.
Minor changes:
1.- Add in the baseline characteristics of the SLE group, type of SLE medication, example corticosteroids, chloroquine, benlista, other IS ....
2.- Add type of dialyzers, time of hemodialysis, type of hemodialysis, Qb, venous and arterial pressure between each group.
3.- Describe in material and methods, type of antihypertensives, dry weight, and interdialytic gain between each group in order to explain the comparisons of SBP and DBP.
4.- Add parameters of residual renal function in each group, such as volume of diuresis, KuR, ClCr, to justify in some way whether this influences the comparative survival in each group.
5.- Add a commentary on the IS medication of the SLE patient group to explain the improvement in survival and mortality relative to the other groups.
Author Response
First of all, we would like to thank the authors for their original and interesting work on the comparison of survival and mortality-related factors among patients undergoing hemodialysis for SLE, DM2 and other causes of CKD in a Korean population.
It is a very interesting manuscript, but it has an important gap on the SLE population studied, although it is commented in the limitations of the study, it would be interesting to describe in some way parameters that identify the severity, activity, time of disease and medication of SLE, probably adding these data in the baseline characteristics of this group.
Answer: Thank you for your comment. Although our study did not include data on the severity, activity, or duration of SLE, we evaluated SLE medications including steroids, hydroxychloroquine, mycophenolate mofetil or mycophenolic acid, cyclophosphamide, azathioprine, and calcineurin inhibitors. The use of medications was defined as one or more prescriptions of a drug during a six-month period of HD quality assessment. The impact of medications for lupus (hydroxychloroquine, steroids [prednisolone, methylprednisolone, deflazacort, azathioprine], mycophenolate mofetil, mycophenolic acid, cyclophosphamide, calcineurin inhibitor [tacrolimus, cyclosporine], rituximab, or belimumab), renin-angiotensin system blockades (RASBs), and anti-hypertensive drugs including RASB, were evaluated.
The number of patients who used steroids, hydroxychloroquine, mycophenolate mofetil or mycophenolic acid, cyclophosphamide, azathioprine, and calcineurin inhibitors was 250 (43.9%), 84 (14.8%), 33 (5.8%), 5 (0.9%), 9 (1.6%), and 22 (3.9%), respectively. However, the number of patients who did not use any of these drugs was 300 (52.7%). Compared to the results of a previous study, the proportion of patients with SLE undergoing HD who did not use immunosuppressive agents (ISAs) was lower [1]. This may indicate that patients with SLE undergoing DM have relatively lower disease activity than those without HD.
Added reference
[1] Han JY, Cho SK, Kim H, Jeon Y, Kang G, Jung SY, Jang EJ, Sung YK. Increased cardiovascular risk in Korean patients with systemic lupus erythematosus: a population-based cohort study. Sci Rep. 2024 Jan 11;14(1):1082.
Minor changes:
1.- Add in the baseline characteristics of the SLE group, type of SLE medication, example corticosteroids, chloroquine, benlista, other IS ....
Answer: Thank you for your comment. We have added the data for medication for SLE and detailed explanations are presented in our answer to the previous comment.
2.- Add type of dialyzers, time of hemodialysis, type of hemodialysis, Qb, venous and arterial pressure between each group.
Answer: Thank you for your comment. Our dataset originated from a HD quality assessment program and their claims data. The purpose of the program was to collect factors associated with improved patient outcomes from a center-based perspective, rather than patient-specific factors linked to individual prognoses. These factors included calcium and phosphorus control, anemia management, and dialysis adequacy. Additionally, due to the nationwide nature of this study, factors that were burdensome to collect or prone to error were excluded. Therefore, our study did not include detailed data on HD sessions, such as changes in blood pressure during HD, HD modalities, blood flow rate during HD, vascular access pressures, dialyzer types, and the duration of HD sessions, which may have served as additional confounding factors affecting patient outcomes.
Nevertheless, in the population-based registry data, some information on HD sessions is available. In South Korea, the use of synthetic membranes, primarily polysulfone, has been increasing and has already exceeded 90% in 2009 [1]. Additionally, in South Korea, reimbursements for HD sessions are provided uniformly, regardless of dialyzer type or dialysis modality. As a result, high-flux dialyzers and HD are predominantly used, while medium-cutoff dialyzers or hemodiafiltration are available only for a limited number of patients. In fact, data from 2018 indicated that only 17% of patients underwent hemodiafiltration at least once per week, and this percentage decreased to 11.1% by 2022, suggesting that the vast majority of HD modalities remain conventional HD. Furthermore, over 80% of patients use dialyzers with surface areas between 1.0 and 2.0 m².
In our dataset, we could identify the number of HD sessions per week. For weekly HD sessions, the proportion of patients undergoing HD twice per week was 48 (8.5%) in the SLE group; 2,074 (8.4%) in the DM group; and 3,613 (11.7%) in the Others group. Although twice-weekly HD sessions were slightly more common in the Others group, the majority of patients in all groups underwent HD thrice times per week.
We have added these comments in the Results and Discussion sections.
Added reference
[1]. ESRD Registry Committee: Korean Society of Nephrology. Current Renal Replacement Therapy in Korea. (assessed 31 December, 2024); https://ksn.or.kr/bbs/index.php?code=report
3.- Describe in material and methods, type of antihypertensives, dry weight, and interdialytic gain between each group in order to explain the comparisons of SBP and DBP.
Answer: Thank you for your comment. We evaluated the use of renin-angiotensin system blockades (RASBs), anti-hypertensive drugs, post-dialysis body weight as an indicator of dry weight, body mass index, and ultrafiltration volume (UFV) as an indicator of weight gain. The post-dialysis body weights in the SLE, DM, and Others groups were 57.4 ± 12.5, 62.7 ± 11.6, and 59.3 ± 11.8 kg, respectively (P < 0.001). Overall, 481 (84.5%), 22,001 (89.2%), and 24,493 (79.0%) patients used anti-hypertensive drugs in the SLE, DM, and Other groups, respectively. The three groups had similar UFV percentages relative to the post-dialysis body weight (approximately 3.7% across the groups). However, the DM group exhibited the highest pre-dialysis SBP and the highest proportion of anti-hypertensive drug or RASB use among the three groups. These results suggest that the higher use of anti-hypertensive drugs, particularly RASBs in patients with DM, may be attributable to a higher prevalence of cardiovascular diseases in this group. Additionally, the elevated pre-dialysis SBP in the DM group may indicate an effort to mitigate the risk of intradialytic hypotension compared to other groups. However, differences in SBP and the use of anti-hypertensive drugs between the SLE and Others groups were not as pronounced as those observed in the DM group.
We have added these comments in the Results section.
4.- Add parameters of residual renal function in each group, such as volume of diuresis, KuR, ClCr, to justify in some way whether this influences the comparative survival in each group.
Answer: Thank you for your comment. As the reviewer has pointed out, residual renal function can be an important factor in patients with HD. Our dataset originated from the HD quality assessment program and their claims data. The purpose of the program was to collect factors associated with improved patient outcomes from a center-based perspective, rather than patient-specific factors linked to individual prognoses. These factors included calcium and phosphorus control, anemia management, and dialysis adequacy. Additionally, due to the nationwide nature of this study, factors that were burdensome to collect or prone to error were excluded. Consequently, data on residual renal function, which requires 24-hour urine collection and is both inconvenient and prone to inaccuracies, were not collected and analyzed in this study. However, most patients included in this study had undergone HD for an average of more than three years. Previous data from a study using the DOPPS cohort showed that 76.1% of patients undergoing HD for more than one year had a urine volume of less than 200 mL/day, and this percentage increased to 83.7% for those undergoing HD for over three years [1]. Furthermore, Japanese patients undergoing HD lost 86.0% of residual renal function after one year and 90.4% after three years. Considering these findings, it is likely that most patients included in our study had minimal or no urine volume.
We have added these comments in the Discussion section.
Added reference
[1] Hecking M, McCullough K, Port FK, et al. Residual urine volume in hemodialysis patients: international trends, predictors and outcomes in the DOPPS [Abstract SA-PO847]. American Society of Nephrology. Kidney Week, October 31−November 5, 2017, New Orleans, LA.
5.- Add a commentary on the IS medication of the SLE patient group to explain the improvement in survival and mortality relative to the other groups.
Answer: Thank you for your comment. We divided patients with SLE into two groups: those who used and those who did not use ISAs. Multivariate Cox regression showed that SLE not treated with ISAs demonstrated a better patient survival than that observed in the DM group (Table S4). However, SLE treated with ISAs exhibited poorer patient survival than that of the Others group, and similar patient survival compared with the DM group. We revealed that patients with SLE had a different patient survival profile, based on the use of ISAs.
Table S4. Cox regression analyses for survival after dividing patients with SLE according to the use of ISAs
|
|
Univariate |
Multivariate |
||
|
HR (95% CI) |
P |
HR (95% CI) |
P |
|
|
Ref: SLE without ISAs |
|
|
|
|
|
SLE with ISAs |
1.04 (0.80–1.36) |
0.783 |
1.44 (0.96–2.15) |
0.075 |
|
Diabetes |
1.89 (1.56–2.28) |
<0.001 |
1.65 (1.21–2.26) |
0.002 |
|
Others |
1.05 (0.87–1.27) |
0.599 |
0.99 (0.73–1.35) |
0.963 |
|
Ref: SLE with ISAs |
|
|
|
|
|
Diabetes |
1.81 (1.51–2.20) |
<0.001 |
1.15 (0.89–1.48) |
0.288 |
|
Others |
1.01 (0.84–1.22) |
0.892 |
0.69 (0.53–0.89) |
0.004 |
The multivariate analysis was adjusted for the disease group, age, sex, CCI score, hemodialysis vintage, ultrafiltration volume, Kt/Vurea, systolic blood pressure, and diastolic blood pressure, as well as levels of hemoglobin, serum albumin, serum creatinine, serum phosphorus, serum calcium; and it was performed using the backward mode. Abbreviations: SLE, systemic lupus erythematosus; ISA, immunosuppressive agent.

Round 2
Reviewer 1 Report
Comments and Suggestions for Authors
the authors expanded the paper by making the requested changes
Reviewer 2 Report
Comments and Suggestions for Authors
The authors have made the suggested changes, improving the clarity of the article. I would accept the version submitted to the journal for publication.